Combined effect of 17β-estradiol and resveratrol against apoptosis induced by interleukin-1β in rat nucleus pulposus cells via PI3K/Akt/caspase-3 pathway

Yang Si-Dong 1
Ma Lei 1
Yang Da-Long 1
Ding Wen-Yuan 2 dingwyster@126.com
1 Department of Spinal Surgery, The Third Hospital of Hebei Medical University , Shijiazhuang, Hebei , People’s Republic of China
2 Department of Spinal Surgery, The Third Hospital of Hebei Medical University; Hebei Provincial Key Laboratory of Orthopaedic Biomechanics , Shijiazhuang, Hebei , People’s Republic of China
Goyal Pankaj
Electronic publication date: 2016 Jan 26
Publication date: 2016
Volume: 4
Electronic Location ID: e1640
Received 2015 Oct 22; Accepted 2016 Jan 8
Copyright: © 2016 Yang et al.
Copyright year: 2016
Copyright holder: Yang et al.
License: This is an open access article distributed under the terms of the Creative Commons Attribution License, which permits unrestricted use, distribution, reproduction and adaptation in any medium and for any purpose provided that it is properly attributed. For attribution, the original author(s), title, publication source (PeerJ) and either DOI or URL of the article must be cited.
License URL: https://creativecommons.org/licenses/by/4.0/

Keywords: Apoptosis, Resveratrol, PI3K/Akt, Intervertebral disc degeneration, Nucleus pulposus, 17β-estradiol

Funding: Natural Science Fund of China 81572166 Natural Science Fund of Hebei Province H2014206075 This study was supported by Natural Science Fund of China (No. 81572166) and Natural Science Fund of Hebei Province (No. H2014206075). The funders had no role in study design, data collection and analysis, decision to publish, or preparation of the manuscript.

==============================
Background: In previous studies, both 17β-estradiol (E2) and resveratrol (RES) were reported to protect intervertebral disc cells against aberrant apoptosis. Given that E2 has a better anti-apoptotic effect with more cancer risk and RES has an anti-apoptotic effect with less cancer risk, the combined use of E2 with RES is promising in developing clinical therapies to treat apoptosis-related diseases such as intervertebral disc degeneration in the future. Objective: The purpose of this study was to explore the combined effect of E2 with RES on rat nucleus pulposus cells and the underlying mechanisms. Methods: TUNEL assay and FACS analysis were used to determine apoptotic incidence of nucleus pulposus cells. MTS assay was used to determine cell viability, and cellular binding assay was used to determine cell-ECM (extracellular matrix) ability. Real-time quantitative RT-PCR was to determine mRNA level of target genes. And Western blot was used to determine the protein level. Results: Both E2 and RES decreased apoptotic incidence when used singly; interestingly, they decreased apoptosis more efficiently when used combinedly. Meanwhile, E2 and RES combined together against the decrease of cell viability and binding ability resulting from IL-1β cytotoxicity. As well, activated caspase-3 was suppressed by the combined effect. Furthermore, IL-1β downregulated expression level of type II collagen and aggrecan (standing for anabolism), while upregulated MMP-3 and MMP-13 (standing for catabolism). However, the combined use of E2 with RES effectively abolished the above negative effects caused by IL-1β, better than either single use. Finally, it turned out to be that E2 and RES combined together against apoptosis via the activation of PI3K/Akt/caspase-3 pathway. Conclusion: This study presented that IL-1β induced aberrant apoptosis, which was efficiently resisted by the combined use of E2 with RES via PI3K/Akt/caspase-3 pathway.

Introduction

Clinically, intervertebral disc (IVD) degeneration (IVDD) is commonly seen, and is often associated with lower back pain. It has been demonstrated in previous studies that aberrant apoptosis and accelerated ageing of nucleus pulposus cells (NPCs) are considered as the two major cellular processes which are closely related to IVDD (Le, Freemont & Hoyland, 2007; Yang et al., 2015). It indicates that NPC apoptosis may be a potential target when we think about prevention and treatment of IVDD.

In our in-vitro studies, 17β-estradiol (E2) has been found to have a protective effect on rat IVD cells (Wang et al., 2014; Yang et al., 2014a; Yang et al., 2015). In addition, E2 has been elucidated to protect against aberrant apoptosis on NPCs via the up-regulation of type II collagen (COL2α1) and aggrecan, and down-regulation of MMP-3 and MMP-13 (Yang et al., 2015). However, estrogen use is often related to more cancer risk. Thus, it limits its further use in clinical aspect. Resveratrol (RES) is a natural polyphenol compound, also known as a phytoestrogen (Li et al., 2008; Zou et al., 2014). In recent years, some in-vivo (Wuertz et al., 2011; Kwon, 2013) and in-vitro studies (Li et al., 2008; Wuertz et al., 2011; Shen et al., 2013) have reported its protective effect on IVD. It was reported that RES significantly protected rabbit discs by decreasing loss of aggrecan and inhibiting MMP-13 level (Kwon, 2013). In addition, RES can effectively restrain inflammatory factor IL-6, IL-8 (Wuertz et al., 2011), suppress catabolism including MMP-1, MMP-3, MMP-13, and ADAMTS-4 (Li et al., 2008; Wuertz et al., 2011; Kwon, 2013), and enhance anabolism of proteoglycan (Li et al., 2008).

Furthermore, E2 (Jover-Mengual et al., 2010; Huang et al., 2011; Ld et al., 2012; Garrido et al., 2013; Mo et al., 2013; Okoh et al., 2013; Qi et al., 2014) and RES (Venkatachalam et al., 2008; Jiang et al., 2009; Roy et al., 2009; Chan et al., 2013; Tsai et al., 2013; Lin et al., 2014; Sui et al., 2014; Chong et al., 2015; Liu et al., 2015) have been widely reported to have a close relationship with PI3K/Akt signal pathway. Also, it has been found that estrogen receptors (ERs, including ER-α and ER-β) are involved in the RES-mediated effects (Robb & Stuart, 2011; Di et al., 2012; Nwachukwu et al., 2014).

Therefore, given that E2 has a better anti-apoptotic effect with more cancer risk and RES has an anti-apoptotic effect with less cancer risk, we have built the hypothesis that the combined use of E2 with RES may potentially play a more efficient role in retarding the progress of IVDD-related diseases. The aim of the present study is to explore the combined effect of E2 with RES on cell apoptosis induced by IL-1β in rat NPCs. In the meanwhile, the role of PI3K/Akt pathway in signal transduction has been preliminarily studied.

Materials and Methods

Ethics statement

Animal protocols were approved by the Institutional Animal Care and Use Committee of The Third Hospital of Hebei Medical University. The approval number is K2015-019-1.

Reagents and antibodies

The information about reagents and antibodies used in this study was collected in Table 1.

Table 1 Reagents and antibodies used in the experiments.

Reagents/antibodies	Manufacturers	City/country	Catalog no.	Source	
FBS	Gibco	New York, US	10099-141	Bovine	
Caspase-3 p17 antibody	Santa Cruz	Santa Cruz, CA	sc-98785	Rabbit	
Akt antibody	Cell signaling	Beverly, MA	2920	Mouse	
p-Akt antibody	Cell signaling	Beverly, MA	12694	Mouse	
GAPDH	Proteintech	Wuhan, China	10494-1-AP	Rabbit	
IL-1β	Peprotech	Rocky Hill, NJ	400-01B	Rat	
Secondary antibodies	Proteintech	Wuhan, China	SA00001	Goat	
Resveratrol	Sigma-Aldrich	St. Louis, MO	R5010	Grape skin	
Trypsin	Sigma-Aldrich	St. Louis, MO	T4049	Porcine	
Collagenase type II	Sigma-Aldrich	St. Louis, MO	C2139	Rat	
17β-estradiol	Sigma-Aldrich	St. Louis, MO	491187	N/A	
DMSO	Solarbio	Beijing, China	D8372-100	N/A	
DMEM/F12	Gibco	New York, US	11320-033	N/A	
TUNEL detection kit	Promega	Madison, WI	G3250	N/A	
MTS Reagent Solution	Promega	Madison, WI	G3582	N/A	
Caspase-3 activity kit	Beyotime	Shanghai, China	C1115	N/A	
LY294002	Selleck Chemicals	Huston, TX	S1105	N/A	
ICI182,780	Selleck Chemicals	Huston, TX	S1191	N/A	
PE Annexin V Apoptosis	BD Pharmingen	San Jose, CA	559763	N/A	
Detection Kit I					

Cell culture

Cell culture was performed according to our previous report (Yang et al., 2015). Briefly, three male Sprague-Dawley rats (∼200 g, 2 months) were sacrificed by anesthetic overdose with pentobarbital sodium. Lumbar spinal columns were removed en bloc under aseptic conditions, and lumbar IVDs were collected. The gel-like nucleus pulposus was separated from the annulus fibrosus under a dissecting microscope and was sequentially treated with 0.25% type II collagenase for 1 hour and 0.2% trypsin with EDTA (1 mmol/L) for 5 minutes. The solution containing partially digested tissue was then transferred to a 50 mL culture flask containing DMEM and 20% FBS and cultured in a humidified atmosphere of 5% CO2 at 37 °C. NPCs were adherent to the bottom of culture flask after 3 days. When confluent (after 1 week), NPCs were dissociated using 0.25% trypsin with EDTA (1 mmol/L) solution and further sub-cultured. The primary cultured cells were seeded in triplicates. First-passage (P1) cells maintained in monolayers were used for the following experiments (n = 6 per group). FBS is known to contain a variety of growth factors and other factors required for cells to survive in culture. It is therefore likely that it would be better to use IL-1β to induce cell apoptosis with serum deprivation. Moreover, phenol red, a pH indicator that has weak estrogenic properties, strongly improved cell proliferation and reduced cell apoptosis (Yang et al., 2014b). Therefore, the presence of phenol red may to some degree disturb the accuracy of experimental data and as a result, reduce the reliability of the whole research. Therefore, all experiments in this study were conducted without phenol red.

Fluorescence-activated cell sorting (FACS) analysis

FACS analysis was performed according to our previous report (Yang et al., 2015). NPCs were plated into 6-well plates at a density of 2 × 105 cells per well and divided into ten groups. As a control, group A was treated with vehicle mixture (ethanol and DMSO, <0.1%; ethanol was the solvent of E2 and DMSO was the solvent of RES). Group B was treated with 75 ng/ml IL-1β (Yang et al., 2015). Group C was treated with 75 ng/ml IL-1β with the pretreatment of 1 μM E2 for 30 min (Yang et al., 2015). Group D was treated with 75 ng/ml IL-1β with the pretreatment of 10 μM RES for 30 min. Group E was treated with 75 ng/ml IL-1β with the pretreatment of 100 μM RES for 30 min. Group F was treated with 75 ng/ml IL-1β with the pretreatment of 200 μM RES for 30 min. Group G was treated with 75 ng/ml IL-1β with the pretreatment of 1 μM E2 and 10 μM RES for 30 min. Group H was treated with 75 ng/ml IL-1β with the pretreatment of 1 μM E2 and 100 μM RES for 30 min. Group I was treated with 75 ng/ml IL-1β with the pretreatment of 1 μM E2 and 200 μM RES for 30 min. Group J was treated with 75 ng/ml IL-1β, 1 μM E2 and 200 μM RES, with the pretreatment of 1 μM ICI182780 for 30 min. For all subsequent experiments, RES was used at a concentration of 200 μM, as this is the maximum concentration found to maintain disc cell viability and provide consistency from experiment to experiment (Li et al., 2008). All groups were incubated for 24 h in the serum-free medium without phenol red. Apoptotic cells were detected using an Annexin V-FITC/PI kit (BD Pharmingen, San Jose, CA, USA) according to the manufacturer’s instructions. Annexin V and PI binding were assessed (within 1 h) using a flow cytometer (BD Biosciences, San Jose, CA, USA) with CellQuest (BD Biosciences, San Jose, CA, USA) software. Apoptotic cells, stained positive for annexin V-FITC, negative for PI, or double positive, were counted. In the experiments, no less than 3,500 cells per sample were analyzed by FACS. In fact, most were more than 5000 cells per sample that were analyzed by FACS. Data are represented as a percentage of the total cell count.

Inverted fluorescence microscopy

NPCs were divided into five groups. As a control, group A was treated with vehicle mixture (ethanol and DMSO, <0.1%; ethanol was the solvent of E2 and DMSO was the solvent of RES). Group B was treated with 75 ng/ml IL-1β. Group C was treated with 75 ng/ml IL-1β with the pretreatment of 1 μM E2 for 30 min. Group D was treated with 75 ng/ml IL-1β with the pretreatment of 200 μM RES for 30 min. Group E was treated with 75 ng/ml IL-1β with the pretreatment of 1 μM E2 and 200 μM RES for 30 min. All groups were incubated for 24 h in the serum-free medium without phenol red. TUNEL assay was performed to detect apoptosis according to manufacturer’s instructions. Apoptotic changes of NPCs were observed under an inverted fluorescence microscope (Olympus, Japan) and photographed by a digital camera (Nikon, Japan). About 200 cells were counted per sample in the analysis of apoptotic changes of NPCs by the Tunel assay.

MTS assay

Cell viability was determined by MTS assay using CellTiter 96® AQueous MTS Reagent Solution (Promega, Madison, WI, USA) according to manufacturer’s instruction. NPCs were treated in five groups as above and then seeded into 96-well plates. The optical density was measured at 492 nm with a microplate reader (Shimadzu, Kyoto, Japan) and cell viability was normalized as a percentage of control.

Cellular binding assay

Rat NPCs were treated in five groups as described above and then assayed for their ability to bind COL2α1 according to the reported method (Yang et al., 2014a). Briefly, a 24-well plates were coated over night at 4 °C with COL2α1 (20 μg/ml; Sigma-Aldrich, St. Louis, MO USA). Non-specific binding sites were blocked by incubating the coated plates with 10 mg/ml albumin for 60 min followed by two washes with ice-cold PBS. A total of 30,000 cells were placed in each well and allowed to adhere at 37 °C for 60 min. After adhesion, cells were stained with 0.5% toluidine blue, fixed with 4% paraformaldehyde, and solubilized in 1% SDS. Extracted dye was quantified by measuring absorbance value at 590 nm in a plate reader (Dynatech MR5000; Dynatech Labs, Chantilly, VA, USA).

Active caspase-3 activity assay

Active caspase-3 activity was performed according to our previous report (Yang et al., 2015). NPCs (2 × 105) were cultured in 6-well plates in five groups as described above. Caspase-3 activity was determined using a caspase-3 activity kit (Beyotime, Shanghai, China), which is based on the ability of caspase-3 to catalyze the formation of a yellow formazan product, p-nitroaniline, from acetyl-Asp-Glu-Val-Asp p-nitroanilide. As recommended by the manufacturer’s protocol, treated cells were lysed with lysis buffer (100 μL per 2 × 106 cells) for 15 min on ice followed by washing with cold PBS. Solutions containing 10 μL cell lysate, 80 μL reaction buffer and 10 μL 2 mM caspase-3 substrate were then incubated in 96-well microtiter plates at 37 °C for 3 h. Caspase-3 activity was quantified with a microplate spectrophotometer (Biotek, Winooski, VT, USA) at an absorbance of 405 nm. Caspase-3 activity is expressed as the fold change in enzyme activity over control.

RT-qPCR

PCR was performed according to our previous report (Yang et al., 2015). The changes of mRNA were detected by RT-qPCR (reverse transcription and real-time quantitative polymerase chain reaction). A total of 10 μg RNA was isolated with Trizol (Solarbio, Beijing, China) according to the manufacturer’s instructions. Total RNA was measured fluorometrically using a CyQuant-Cell Proliferation Assay Kit (Molecular Probes, Eugene, OR, USA). cDNA synthesis was performed using a ThermoScript RT-qPCR System (Invitrogen, Shanghai, China). For quantification of genes of interest we utilized a DyNAmo SYBR Green 2-step RT-qPCR Kit (Promega, Madison, WI, USA) in a total volume of 20 L, with real-time PCR performed using an Mx300P cycler. Amplicons of target genes were amplified with the primers designed with Primer Premier Version 5.0 software and their efficiency was confirmed by sequencing their conventional PCR products (Table 2). PCR amplification was performed using the following protocol: 95 °C for 2 min, then 40 cycles of 95 °C for 15 sec, and finally 60 °C for 1 min. Standard curves were generated for each assay to produce a linear plot of threshold cycle (Ct) against log (dilution). Target gene expression was quantified according to the concentration of the standard curve. Data are presented as relative Ct values (n = 6). Relative gene expression data was calculated according to the 2−ΔΔCt method. Target gene expression was normalized to reference gene (GAPDH).

Table 2 Primers used in the study.

No.	Target gene	Sequence (sense, anti-sense, 5′-3′)	Accession no. (GenBank)	
1	COL2α−1	ACGCTCAAGTCGCTGAACAA
TCAATCCAGTAGTCTCCGCTCT	NM_012929	
2	Aggrecan	TCCAAACCAACCCGACAAT
TCTCATAGCGATCTTTCTTCTGC	NM_022190	
3	MMP-3	ATGATGAACGATGGACAGATGA
CATTGGCTGAGTGAAAGAGACC	NM_133523	
4	MMP-13	GGCCAGAACTTCCCAACCA
ACCCTCCATAATGTCATACCC	NM_133530	
5	GAPDH	GGAAAGCTGTGGCGTGAT
AAGGTGGAAGAATGGGAGTT	NM_017008	

Western blot

Western blot was performed according to our previous report (Yang et al., 2015). Protein levels of Akt, p-Akt (Ser473) and active caspase-3 was determined by western blot, with GAPDH as internal reference protein. Rat NPCs were washed with ice-cold PBS and harvested in 100 μL of cell Lysis buffer containing 1% protease inhibitor (Solarbio, Beijing, China). Lysates were centrifuged at 4 °C for 5 min at 14,000 rpm and resolved on 12% SDS-polyacrylamide gels (SDS-PAGE). Proteins were transferred by electroblotting to a PVDF membrane (Merk Millipore, Billerica, MA, USA). The membranes were blocked with 5% non-fat dry milk in TBS (50 mmol/L Tris, pH 7.6, 150 mmol/L NaCl, 0.1%) and incubated overnight at 4 °C in 5% non-fat dry milk in PBST with corresponding first antibodies (dilution 1:500). Washing in PBST 3 times for 30 min, the membrane was incubated with a secondary anti-IgG-HRP antibody (dilution 1:5000) at room temperature for 2 h. Immunolabeling was detected using enhanced chemiluminescence (ECL) reagent (Solarbio, Beijing, China).

Statistical analysis

Statistical analyses were performed using SPSS for Windows, version 18.0 (SPSS Inc., USA). All data are presented as the mean ± SD (standard deviation) of independent experiments performed (n = 6 for each group). If data satisfied criteria for normality and homogeneity of variance, statistical analysis among multiple groups was performed using one-way analysis of variance (ANOVA), accompanied by pairwise comparison using the SNK-q test. If not, statistical analysis was performed using non-parametric rank test. p < 0.05 was regarded as statistically significant.

Results

FACS analysis

As shown in Figs. 1A and 1B, IL-1β resulted in a marked increase (up to 14%) of apoptotic incidence. However, the apoptotic incidence induced by IL-1β could be effectively decreased by the single use of 1 μM E2 or 200 μM RES, as well as the combined use of 1 μM E2 and 200 μM RES.

Figure 1 FACS analysis for apoptotic incidence.

(A) NPCs were plated into 6-well plates at a density of 2 × 105 cells per well and divided into ten groups as presented in the figure. All groups were incubated for 24 h in the serum-free medium without phenol red. Apoptotic cells were detected using an Annexin V-FITC/PI kit (BD Pharmingen, San Jose, CA, USA) according to the manufacturer’s instructions. Apoptotic cells, stained positive for annexin V-FITC, negative for PI, or double positive, were counted. (B) Data are represented as a percentage of the total cell count. Data analysis was determined by one-way analysis of variance (ANOVA) accompanied by pairwise comparison using SNK-q test. * p < 0.05 by one-way analysis of variance (ANOVA) accompanied by pairwise comparison using SNK-q test. NPCs, nucleus pulposus cells; IL-1β, interleukin-1β; E2, 17β-estradiol; RSV, resveratrol; mean ± SD (standard deviation); n = 6.

Inverted fluorescence microscopy

As shown in Figs. 2A and 2B, TUNEL assay showed that IL-1β induced marked apoptosis (14%) compared to control (7%), which was effectively reversed by the combined use of 1 μM E2 and 200 μM RES (4.5%).

Figure 2 TUNEL assay for apoptosis.

NPCs were divided into five groups. As a control, group A was treated with vehicle mixture (ethanol and DMSO, <0.1%; ethanol was the solvent of E2 and DMSO was the solvent of RSV). Group B was treated with 75 ng/ml IL-1β. Group C was treated with 75 ng/ml IL-1β with the pretreatment of 1 μM E2 for 30 min. Group D was treated with 75 ng/ml IL-1β with the pretreatment of 200 μM RSV for 30 min. Group E was treated with 75 ng/ml IL-1β with the pretreatment of 1 μM E2 and 200 μM RSV for 30 min. All groups were incubated for 24 h in the serum-free medium without phenol red. All cells were stained red by PI, and apoptotic cells presented green. (A) Scale bar, 200 μm; (B) Scale bar, 50 μm; NPCs, nucleus pulposus cells; IL-1β, interleukin-1β; E2, 17β-estradiol; RSV, resveratrol; PI, prodium iodide; n = 6.

MTS assay,cellular binding and active caspase-3 activity

As shown in Figs. 3, 4 and 5, IL-1β resulted in a decrease of nearly 30% both in cell viability (p < 0.05) and cell binding ability (p < 0.05), as well as an increase of 4 fold in activated caspase-3 activity (p < 0.05). However, the cytotoxic effects of IL-1β were partly abolished by the addition of E2 or RES, but all were reversed by the combined use of 1 μM E2 and 200 μM RES (p < 0.05).

Figure 3 MTS assay for cell viability.

NPCs were treated in five groups as presented in the figure and then seeded into 96-well plates. All groups were incubated for 24 h in the serum-free medium without phenol red. Cell viability was determined by MTS assay using CellTiter 96® AQueous MTS Reagent Solution (Promega, Madison, WI, USA) according to manufacturer’s instruction. The optical density was measured at 492 nm with a microplate reader (Shimadzu, Kyoto, Japan) and cell viability was normalized as a percentage of control. * p < 0.05, by one-way analysis of variance (ANOVA) accompanied by pairwise comparison using SNK-q test. NPCs, nucleus pulposus cells; IL-1β, interleukin-1β; E2, 17β-estradiol; RSV, resveratrol; mean ± SD (standard deviation); n = 6.

Figure 4 Cellular binding ability to type II collagen.

NPCs were divided into five groups. As a control, group A was treated with vehicle mixture (ethanol and DMSO, <0.1%; ethanol was the solvent of E2 and DMSO was the solvent of RSV). Group B was treated with 75 ng/ml IL-1β. Group C was treated with 75 ng/ml IL-1β with the pretreatment of 1 μM E2 for 30 min. Group D was treated with 75 ng/ml IL-1β with the pretreatment of 200 μM RSV for 30 min. Group E was treated with 75 ng/ml IL-1β with the pretreatment of 1 μM E2 and 200 μM RSV for 30 min. All groups were incubated for 24 h in the serum-free medium without phenol red. * p < 0.05, by one-way analysis of variance (ANOVA) accompanied by pairwise comparison using SNK-q test. NPCs, nucleus pulposus cells; IL-1β, interleukin-1β; E2, 17β-estradiol; RSV, resveratrol; mean ± SD (standard deviation); n = 6.

Figure 5 Active caspase-3 activity assay.

NPCs were divided into five groups. As a control, group A was treated with vehicle mixture (ethanol and DMSO, <0.1%; ethanol was the solvent of E2 and DMSO was the solvent of RSV). Group B was treated with 75 ng/ml IL-1β. Group C was treated with 75 ng/ml IL-1β with the pretreatment of 1 μM E2 for 30 min. Group D was treated with 75 ng/ml IL-1β with the pretreatment of 200 μM RSV for 30 min. Group E was treated with 75 ng/ml IL-1β with the pretreatment of 1 μM E2 and 200 μM RSV for 30 min. All groups were incubated for 24 h in the serum-free medium without phenol red. Caspase-3 activity was determined using a caspase-3 activity kit (Beyotime, Shanghai, China). Caspase-3 activity is expressed as the fold change in enzyme activity over control. * p < 0.05, by one-way analysis of variance (ANOVA) accompanied by pairwise comparison using SNK-q test. NPCs, nucleus pulposus cells; IL-1β, interleukin-1β; E2, 17β-estradiol; RSV, resveratrol; mean ± SD (standard deviation); n = 6.

RT-qPCR

As shown in Fig. 6, IL-1β significantly reduced 30% COL2α1 and 40% aggrecan, while increased 4 fold of MMP-3 and 5 fold of MMP-13, as compared to control (all p < 0.05). Of note, the combined use of 1 μM E2 and 200 μM RES increased 25% COL2α1 and 30% aggrecan, while decreased 3.5-fold MMP-3 and 4.5-fold MMP-13, as compared to IL-1β group (all p < 0.05).

Figure 6 RT-qPCR analysis.

NPCs were divided into five groups. As a control, group A was treated with vehicle mixture (ethanol and DMSO, <0.1%; ethanol was the solvent of E2 and DMSO was the solvent of RSV). Group B was treated with 75 ng/ml IL-1β. Group C was treated with 75 ng/ml IL-1β with the pretreatment of 1 μM E2 for 30 min. Group D was treated with 75 ng/ml IL-1β with the pretreatment of 200 μM RSV for 30 min. Group E was treated with 75 ng/ml IL-1β with the pretreatment of 1 μM E2 and 200 μM RSV for 30 min. All groups were incubated for 24 h in the serum-free medium without phenol red. * p < 0.05, by one-way analysis of variance (ANOVA) accompanied by pairwise comparison using SNK-q test. NPCs, nucleus pulposus cells; IL-1β, interleukin-1β; E2, 17β-estradiol; RSV, resveratrol; RT-qPCR, reverse transcription and real-time quantitative polymerase chain reaction; MMP, matrix metalloproteinase; mean ± SD (standard deviation); n = 6.

Western blot

As shown in Fig. 7, relative value of p-Akt/GAPDH was up-regulated to 0.75 by the combined use of E2 and RES, while it was down-regulated to 0.15 by the addition of ER antagonist (ICI), or LY294002 (a PI3K/Akt inhibitor). Besides, cleaved caspase-3 was upregulated by IL-1β, but then downregulated by the addition of E2 and RES (p < 0.05).

Figure 7 Protein levels of Akt, p-Akt(Ser473) and active caspase-3.

NPCs were divided into four groups. Group A was treated with 75 ng/ml IL-1β. Group B was treated with 75 ng/ml IL-1β with the pretreatment of 1 μM E2 and 200 μM RSV for 30 min. Group C was treated with 75 ng/ml IL-1β with the pretreatment of 1 μM E2, 200 μM RSV, and 1 μM ICI for 30 min. Group D was treated with 75 ng/ml IL-1β with the pretreatment of 1 μM E2, 200 μM RSV and 50 μM LY294002(PI3K/Akt inhibitor). All groups were incubated for 24 h in the serum-free medium without phenol red. * p < 0.05, by one-way analysis of variance (ANOVA) accompanied by pairwise comparison using SNK-q test. NPCs, nucleus pulposus cells; IL-1β, interleukin-1β; E2, 17β-estradiol; ICI, ICI182780; RSV, resveratrol; LY, LY294002, 50 μM; mean ± SD (standard deviation); n = 6.

Discussion

In the present study, data show that the gene expression levels of MMP-3 and MMP-13 are both increased while the levels of type II collagen and aggrecan are decreased, due to cytotoxic effect of IL-1β. It is notable that the combined use of E2 and RVS has markedly decreased the cytotoxic effect of IL-1β on NPCs, which was presented as the down-regulation of catabolism (the decreased levels of MMP-3 and MMP-13), and the upregulation of anabolism (the increased levels of COL2α1 and aggrecan). Although the single use of E2 or RES can inhibit the catabolism due to cytotoxic effect of IL-1β, obviously, the combined regulation of E2 with RVS is more effective.

As compared to RES, E2 exerts a better effect to reverse the adverse regulation caused by IL-1β, which is indicated by FACS analysis that more apoptotic incidence is decreased. As well, cellular viability and cell binding is restored more in E2 group than those in RES group. Meanwhile, more catabolism due to cytotoxic effect of IL-1β is prohibited by E2 than RES, with more anabolism increased by E2. However, there is no doubt in this study that the combined use of E2 with RES has an advantage over the single use of E2 or RES. Also E2 combined with RES protects rat NPCs against apoptosis, characterized by enhanced biosynthesis of COL2α1 and aggrecan, as well as increased cell viability and binding ability.

It has long been recognized that there is a continuous increase in cell death during human IVDD. Additionally, nutrient supply to the IVD deteriorates during disc degeneration in humans (Yang et al., 2014a). However, it should be noted in the present study that the cytokine IL-1β is sufficient to induce apoptosis of NPCs based on serum deprivation. Indeed, previous studies (Zhao et al., 2007; Yang et al., 2014a) demonstrate that although suppression of apoptosis increases IVD cell survival, IVD cells inevitably die without an adequate supply of nutrients. Thus, improving the IVD nutrient supply may be beneficial for suppression of IVD cell apoptosis.

It is well-known that estrogen has been closely related to cancer risk, especially breast cancer. Different from E2, as a polyphenolic compound, RVS has anti-inflammatory, antioxidant, neuroprotective properties and acts as a chemopreventive agent (Jiang et al., 2009). RVS causes cell cycle arrest and induces apoptotic cell death in various types of cancer cells (Jiang et al., 2009; Tsai et al., 2013). Contrary to the situation in cancer cells, RVS has a protective effect on intervertebral disc cells from apoptosis (Li et al., 2008), as the same in the current study. However, the anti-apoptotic effect of RVS is a little poorer than E2. Given that E2 has a better anti-apoptotic effect with more cancer risk and RES has an anti-apoptotic effect with less cancer risk, the combined use of E2 with RES is promising in developing clinical therapies to treat apoptosis-related diseases such as IVDD in the future.

RVS has beneficial effects on aging, inflammation and metabolism, which are thought to result from activation of the lysine deacetylase, sirtuin 1 (SIRT1), the cAMP pathway, or AMP-activated protein kinase (Nwachukwu et al., 2014). To our knowledge, both E2 (Jover-Mengual et al., 2010; Huang et al., 2011; Ld et al., 2012; Garrido et al., 2013; Mo et al., 2013; Okoh et al., 2013; Qi et al., 2014) and RES (Venkatachalam et al., 2008; Jiang et al., 2009; Roy et al., 2009; Chan et al., 2013; Tsai et al., 2013; Lin et al., 2014; Sui et al., 2014; Chong et al., 2015; Liu et al., 2015) have been widely reported to have a close relationship with PI3K/Akt signal pathway. Also, it has been found that ERs, including ER-α and ER-β, are involved in the RES-mediated effects (Robb & Stuart, 2011; Di et al., 2012; Nwachukwu et al., 2014). In the present study, a preliminary exploration has been performed to detect the involvement of PI3K/Akt pathway in signal transduction arising in the anti-apoptotic process of E2 combined with RES. It was noted that E2 combined with RES activated and upregulated the PI3K/Akt signal pathway, which could be inhibited by LY, as well as ICI. Hence, these results above further confirmed that E2 and RES exert their effects based on the same receptors. However, the relationship of E2 and RES may change in different situations, especially influenced by the dose of RES, as RES may act as an agonist (Macpherson & Matthews, 2010) or an antagonist to compete with E2 for binding to ER-α (Chakraborty, Levenson & Biswas, 2013). That is to say, RES has partial antagonist and partial agonist actions on ER-α (Chakraborty, Levenson & Biswas, 2013). Up to now, the ER subtypes that RES interacts with are controversial, with regard to ER-α (Macpherson & Matthews, 2010; Di et al., 2012; Kang et al., 2013; Nwachukwu et al., 2014) or ER-β (Jackson, Greiwe & Schwen, 2011; Robb & Stuart, 2011; Di et al., 2012; Chen & Chien, 2014). As for the present study, RES acts as an agonist to enhance the cytoprotective effect of E2.

To our knowledge, the potential benefit of RES mainly results from its antioxidant effect (Gueguen et al., 2015) and the interaction with SIRT (Kaeberlein et al., 2005). Oxidative stress arises from a shift in balance that favors the generation of oxygen-derived reactive oxygen species (ROS) over certain antioxidant defense mechanisms (Meng et al., 2015). ROS can induce peroxidation of lipid, leading to a loss of membrane integrity, reduction of mitochondrial membrane potential, and increased permeability to Ca2+ in the plasma membrane (Wassmann, Wassmann & Nickenig, 2004). Mitochondria are key targets of RES. Some previous studies have found, RES modulates mitochondrial ROS production, mitochondrial biogenesis (Kitada et al., 2011) via its interaction with SIRT1 (Bastin, Lopes-Costa & Djouadi, 2011) and energy metabolism via either transcriptional (Zhou et al., 2014) or enzymatic activation of SIRT3 (Chen et al., 2013). With regard to the current work, it is just a preliminary study on the role of PI3K/Akt pathway in signal transduction. Therefore, there are no experiments performed to explore the mitochondrial pathway or oxidative stress. Surely, this is the limitation of our work, which is designed to be explored in a further study.

Up to now, no study has reported any differences between men and women in response to the suggested treatment including E2 in combination with RES, to treat IVDD-related diseases. As we know, women are more sensitive to estrogen deficiency. Thus, this treatment may be more effective to women than men. Surely, when seeking for the potential of E2 in combination with RES, we have to face the possible side effects. One of them is always associated with breast cancer caused by much estrogen use. However, RES has been shown to be a potential protective agent against cancer, inflammatory lesions, diabetes mellitus, and cardiovascular abnormalities. (Baur & Sinclair, 2006; Zou et al., 2014) Hence, the side effect of E2 in combination with RES is unclear regarding their oncogenicity, due to their opposite effects. Further studies on their side effects are needed.

The obtained results from this study were in vitro results. In order to draw any conclusions about the in vivo situation, animal experiments needs to be performed since results from in vitro models cannot always be compared and/or applied to the in vivo situation. Thus, the results in this study suggest a role in vivo of the investigated combined anti-apoptotic effects via PI3K/Akt/caspase-3 signaling pathway of E2 and RES in IVDD. This research provides a novel insight into the anti-apoptotic effect of E2 combined with RES, potentially leading to a better understanding of clinical therapies based on apoptosis, especially to retard the progress of IVDD.

Supplemental Information

Supplemental Information 1 Raw data and histograms.

Raw data and histograms used in the study.

Click here for additional data file.

Additional Information and Declarations

Competing Interests

Author Contributions

Animal Ethics

Data Deposition

The authors declare that they have no competing interests.

Si-Dong Yang performed the experiments, analyzed the data, wrote the paper, prepared figures and/or tables, reviewed drafts of the paper.

Lei Ma performed the experiments, contributed reagents/materials/analysis tools, wrote the paper, prepared figures and/or tables.

Da-Long Yang performed the experiments, contributed reagents/materials/analysis tools, prepared figures and/or tables.

Wen-Yuan Ding conceived and designed the experiments, reviewed drafts of the paper.

The following information was supplied relating to ethical approvals (i.e., approving body and any reference numbers):

1. Institutional Animal Care and Use Committee of The Third Hospital of Hebei Medical University.

2. Animal protocols were approved by the Institutional Animal Care and Use Committee of The Third Hospital of Hebei Medical University. The approval number is K2015-019-1.

The following information was supplied regarding data availability:

All raw data generated in the research have been included in the figures and tables in the manuscript.

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
