# Peer review of "Combined effect of 17β-estradiol and resveratrol against apoptosis induced by interleukin-1β in rat nucleus pulposus cells via PI3K/Akt/caspase-3 pathway"

_PeerJ, doi:10.7717/peerj.1640_

## Round 0.1 · original submission · Major Revisions

Please fully address the concerns and comments of the two expert reviewers.

In addition, I have a serious concern regarding the figure 3(MTS assay for cell viability) and figure 4 ( Cellular binding ability to type II collagen.). It seems that both figures have same numeric values and identical. If authors are taking relative values how the control (vehicle) bar must have SD value "zero"?

·

Basic reporting

Some acronyms need to be better spell out and further revised.
For example, RSV may induce readers to misleading as Rous Sarcoma Virus, rather than resveratrol...

The relationship between qualitative test for apoptosis and the combined effect of estrogens and resveratrol has not been properly addressed. Synergistici mechanism at which level?
ER-receptors?

Experimental design

RT-qPCR of some genes involved in the apoptotic pathways should be performed.
The Authors are invited to investigate the role of mitochondrial distress and/or ER stress (UPR response), at least in the Discussion section, as resveratrol is a phytoestrogen and particularly an anti-oxidant molecule.
The role of oxidative stress has never been addressed in the paper

Validity of the findings

They do not bear any real novelty in the field

Additional comments

The paper may be accepted if thoroughly re-written and equipped with the many experimental recommendations raised by this Reviewer.

Reviewer 2 ·

Basic reporting

Yang et al. report that 17β-estradiol and resveratrol have synergistic effects against IL-1β-induced apoptosis in rat nucleus pulposus cells. Caspase-3 activation was suppressed by combination of both substances. Anabolism proteins collagen type II and aggrecan were downregulated whereas the catabolism involved proteins MMP-3 and MMP-13 were upregulated. The suppression of apoptosis was regulated via activation of PI3K/AKT/Caspase 3 pathway. This finding might be of relevance for treatment of apoptosis-related diseases like intervertebral disc generation.
There are a lot of typing mistake throughout the whole manuscript, especially blanks are missing (e.g. but not limited to line 40, 65, 72, 75, 77, 98, 108, 118, 160,…). Manuscript has to be carefully read and these mistakes have to be corrected. Terms like “in vitro” or “in vivo” have to be in italic (line 72/73, 304, 305, 307

Experimental design

Generally good and valid methods are used.
But for determination of protein levels of Akt, pAkt and active Caspase 3 as well as GAPDH (Figure 9), I am missing any kind of solvent control to show the basal protein level of the respective proteins.
Cellular binding assay: apart from citing a previous work, at least a brief overview about the method should be given.
For transparence, antibody catalogue numbers and source should be given, and in Table 1 it has to be “secondary antibodies”, not “second antibodies”.

Validity of the findings

Figure 1A: the figure seems to be doubtful as there is a discrepancy in between mat/meth part for the labeling and the figure labeling giving a feeling like wrong data are shown. According to mat/meth part, the figures should be labelled from A to I, but in fact, they are labelled B, C, C, D, B, E, F, F, G. This gives an impression that pictures from the wrong experimental setup are shown. Carefully check out if the correct pictures are shown – if yes, change the labeling, if no – replace by the correct ones.
Apart from that, there should be only one figure legend where everything is embedded with the marking A) and B) for the specific explanations whereas general explanations valid for the respective sub-figures should be in the general text.
Figure 1 B, 3, 4, 5, 7: do really all calculations have a *** significance? Or are there differences in the significance levels? It is hard to believe that independent of the differences in calculated values and their standard deviations that all significance levels should be the same.
Figure 2A and 2B: what is the actual difference of these two figures apart from figure 2A showing the single pictures as well as the merged picture? According to the figure legend, it is in both figures the TUNEL assay which is shown with two different magnifications; the whole text is basically identical with the exception of the scale bar. When using figure labeling A and B, you should only write one figure legend and mark the specificities by writing A) scale bar… and B) scale bar… rather than repeating the whole text. In my opinion, it would be better to show first the overview picture (scale bar 200 µm) and then the magnification (scale bar 50 µm) as part of the overview picture.
Figure 6: what does the axis labeling “relative value of gene expression” mean? And why are the p-values instead of stars shown? It is for sure a valid method to show data, but it should be consistent throughout the whole manuscript.
For me, it is also surprising that for adhesion, proliferation and Col2α1 expression look literally the same with only very slight deviations.

Additional comments

Data are interesting and might be helpful in future for patients.
However, manuscript has to be improved to sort out doubtful data to publish it.

---

## Round 0.2 · Minor Revisions

The manuscript may be accepted after the remaining minor changes.

Reviewer 2 ·

Basic reporting

Just a few technical comments:
Line 41: “Western” has to be in capital.
Line 324: the studies “were” in vitro studies.
Line 73/74: I would put the reference at the end of the sentence.
Methods RT-qPCR: What was the number of cycles? And the conditions for the respective PCS steps? According to MIQE guidelines you should mention both.
Statistics: According to common nomenclature, * indicates p<0.05; ** indicates p<0.01; *** indicates p<0.001, marking should be accordingly.

Experimental design

no comments.

Validity of the findings

Concerning Figure 1A: I would still strongly suggest you to either change the letters in the pictures or remove them as every reader might have the same doubtful impression like me that the wrong pictures are taken. Various programs exist to edit even pdf-files, but not every reader will ignore the discrepancy of text and figures due to a simple mistake of randomly given letters and a lot of doubt might be on the data.

---

## Round 0.3 · accepted · Accept

The manuscript is now suitable for publication in PeerJ.